# Would Surgeons Like to Be Submitted to Anal Fistulotomy? An International Web-Based Survey

**DOI:** 10.3390/jcm12030825

**Published:** 2023-01-20

**Authors:** Carlo Ratto, Angelo Alessandro Marra, Angelo Parello, Veronica De Simone, Paola Campennì, Francesco Litta

**Affiliations:** 1Proctology Unit, Fondazione Policlinico Universitario “Agostino Gemelli” IRCCS, 00168 Rome, Italy; 2Department of Medicine and Translational Surgery, Università Cattolica del Sacro Cuore, 00168 Rome, Italy

**Keywords:** anal fistula, anal fistula surgery, fistulotomy, online survey, participation rate, quality of life, anal continence impairment

## Abstract

Traditional fistulotomy is the most performed surgical procedure in anal fistula surgery. We conducted an international online survey to explore colorectal surgeons’ opinions and preferences on fistulotomy. Considering the healing and continence impairment rates reported in the literature, surgeons were invited to answer as a hypothetic patient susceptible to being submitted to fistulotomy for low and high anal fistula. A total of 767 surgeons completed the survey from 72 countries. The majority of respondents were consultants, having treated more than 20 anal fistulas in the last year. Most of them declared that anal fistula would be able to negatively affect quality of life and would be worried/anxious about it. Taking into account all aspects, 87.5% and 37.8% of respondents would agree to be treated with a fistulotomy in case of a low and high fistula, respectively, with an acceptance rate that varied worldwide. At multivariate analysis, factors correlated to the acceptance of anal fistulotomy were male gender (*p* = 0.003), practice of less than 20 fistula operations during last year (*p* = 0.020), and low fistula (*p* < 0.001). Surgeons recognized the extreme complexity of this approach. This study highlighted the necessity of an accurate patients’ selection and the adoption of alternative strategy to reduce the risk of anal continence impairment.

## 1. Introduction

Anal fistula is a debilitating disease with an incidence of 68,000–96,000 cases per years in the US [1] and a prevalence of 1.2 to 2.8/10,000 in the Europe [2]. This disease has afflicted patients and challenged surgeons since the time of Hippocrates [3]. Traditionally, Parks’ classification has been used in anal fistula surgery to describe the relationship between the fistula tract and anal sphincters [4]. However, a simplified classification in “low” and “high” anal fistula is widely accepted, according to the amount of involved sphincter complex [5,6,7].

Surgery remains the main therapeutic option for anal fistulas, in a delicate balance between the healing of the anal sepsis and preservation of the anal continence [8]. Traditional fistulotomy or “lay-open” technique is still the most performed surgical operation in the world [9]. Although optimal healing rates were shown in 85% to 98% of patients undergoing fistulotomy, some degree of postoperative continence impairment was reported in 6–28% and 17.5–40% cases of low and high anal fistula, respectively [7,10,11,12,13,14,15,16,17,18,19,20,21,22]. Therefore, several alternative sphincter-saving approaches were developed, reporting variable, disappointing success rates [23,24,25,26,27].

Surgeons usually require the patient’s agreement to a surgical treatment that should take in account so many technical factors whose the patient frequently could be not fully conscious. On the other hand, surgeons performing fistulotomy in their daily practice should be perfectly conscious of the various factors related to this procedure. The aim of this study was to investigate surgeons’ opinions about a hypothetic indication to fistulotomy for themselves: they were asked to simulate being a potential patient affected by either a low or high anal fistula and, for this reason, susceptible to being submitted to an anal fistulotomy.

## 2. Materials and Methods

### 2.1. Study Design and Participants

An international online survey was promoted by the Proctology Unit of the Fondazione Policlinico Universitario Agostino Gemelli IRCCS, Rome, Italy to explore surgeon’s opinion and preference about fistulotomy in the management of either low or high anal fistula. Questionnaire respondents were surgeons and trainees from colorectal or general surgery units with practice in fistulotomy across Europe, Africa, Asia, Oceania, North and Middle–South America. Participation to the survey was voluntary, no rewards were offered. This study followed the Checklist for Reporting Results of Internet E-Surveys (CHERRIES statement) [28] and was approved by our local ethical committee.

### 2.2. Survey Development and Content

After a discussion on the topics and questions, the questionnaire was developed using an online platform, the “Online Surveys (formerly Bristol Online Survey—BOS)”, provided by JISC at https://www.onlinesurveys.ac.uk (accessed on 1 June 2020) and certified to ISO 27001 standard. The survey was designed reducing at the best complexity and number of questions. All questions were set as mandatory fields with automated skip logic to avoid missing data. Its technical functionality was tested before the invitation link was sent. The estimated time needed to complete the questionnaire was 5–6 min.

As premise of the questionnaire, it was clarified that, according to the Clinical Practice Guidelines Committee of the American Society of Colon and Rectal Surgeons [6], the anal fistula located at the level of the low anal canal, with less than 30% of anal sphincters’ total length being below the fistula was considered “low fistula”, while the fistula located at the level of the medium-high anal canal, with more than 30% of anal sphincters total length being below the fistula was considered “high fistula”. It was summarized that, even if the high healing rates following fistulotomy in both low and high anal fistulas has been very well-documented (ranging from 85% to 98% in the literature), controversies still exist about the right indications for fistulotomy, mainly due to the possible risk of any continence impairment to feces following this operation (ranging from 6% to 28% in low anal fistulas, and from 17.5% to 40% in high anal fistulas) [7,10,11,12,13,14,15,16,17,18,19,20,21,22]. In this regard, it was underlined that not only the occurrence of severe fecal incontinence (affecting the minority of patients), but also soiling and inadvertent leakage of gas (more frequent) can affect the patients’ quality of life, limiting, reducing, or modifying their daily activities.

This questionnaire was composed of four sections including a total number of 19 questions. The complete survey is available as a Appendix A of this article (Appendix A). Baseline information about respondents such as gender, age group, country of practice, type of hospital (academic, non-academic teaching, non-teaching), and training level (consultant, resident, fellow) were collected in the first section. The second section investigated participants’ experience in surgical management of anal fistula. In the last two sections, respondents assumed to be a patient with either a low or high anal fistula and described psychological and surgical aspects of the fistula disease.

### 2.3. Study Circulation

The questionnaire was made available online for two months. A link to the survey was directly sent by a personal email invitation to experts in colorectal surgery who performed anal fistula procedures. Furthermore, another link to the survey was shared on major social media (LinkedIn, Twitter, Facebook), addressing the main colorectal scientific societies, such as the American Society of Colon and Rectal Surgery, European Society of Coloproctology, Italian Society of Colorectal Surgeons, Russian School of Coloproctology, Brazilian Society of Coloproctology, Mexican College of Coloproctology, and the Colorectal Surgical Society of Australia and New Zealand. Agreeing to participate in the survey, the surgeons could enter their personal email address, where they received the link to the questionnaire. The detailed aim of this study was elucidated to participants in the first page of the online survey. Once participation had been accepted, participants were identified as unique visitors through their personal email in order to avoid duplicate entries.

### 2.4. Data Collection and Analysis

Participation rate (defined as the number of surgeons who accepted to participate in the first page of the survey divided by the number of visitors who visited the first survey page) was used as response metrics. Completed questionnaires were automatically collected in the software and exported to a Microsoft Excel spreadsheet for statistical analysis. The survey data were reported using frequencies and percentages. Categorical variables were assessed with a chi-squared test. Logistic regression model was used for uni- and multivariate analysis.

A *p*-value less than 0.05 was considered statistically significant. Data analysis was performed using IBM SPSS Statistics for Windows, version 25.0 (IBM Corp, Armonk, NY, USA).

## 3. Results

Overall, 985 surgeons accepted to participate in the survey; 767 surgeons completed the survey from 72 countries, with a participation rate of 77.9%. Respondents’ characteristics are summarized in Table 1. Most surgeons came from Europe (the three most representative countries were Italy, Russia, and Spain), North America (mostly from USA), and Middle–South America (the three most representative countries were Mexico, Brazil, and Argentina) (Appendix A). The affiliation to an academic hospital was declared by 55.1% of respondents. Although a few residents and fellows participated to the survey (5.0% and 6.0%, respectively), the majority of respondents were consultants (89.0%). Two thirds of surgeons (66.6%) had more than 10 years of experience in anal fistula management. More than 60% of respondents had treated more than 20 anal fistula patients in the last year.

Figure 1 reports the questionnaire results concerning low and high anal fistulas. Majority of respondents felt that the presence of the anal fistula would be able to affect (either very much or much) their quality of life negatively, limiting/reducing/modifying their daily activities. This feeling was reported by 68.6% of respondents for low fistulas and 88.2% of them concerning high fistulas: the difference was statistically significant (*p* < 0.001). Most of survey participants declared they would be very much or much worried/anxious because of an anal fistula, although with a significant difference between high and low fistulas (87.9% versus 65.4%, respectively, *p* < 0.001). Regarding the perspectives of healing and failure following fistulotomy, 81.4% and 61.5% of respondents considered these perspectives absolutely or enough acceptable for a low fistula and high fistula, respectively (*p* < 0.001). The possibility to be submitted to more than one surgical procedure to treat an anal fistula worried much/very much 62.7% and 70.5% of participants for low and high fistula, respectively (*p* < 0.001). For low fistulas, the risk of continence impairment (as reported by the literature) was considered as absolutely or enough acceptable by 34.6% of the respondents, still considerable by 39.8%, and not too much or not at all acceptable by 25.6%. Concerning high fistulas, responses to this question were 19.3%, 31.1%, and 49.6%, respectively. The final question investigated the surgeons’ agreement to be treated with a fistulotomy taking into account all the aspects concerning this procedure. Having a low fistula, 87.5% of respondents would agree, having a high fistula, 37.8% of them would agree (*p* < 0.001).

Factors significantly correlated to the agreement to be submitted to fistulotomy were male gender (*p* = 0.003), to perform less than 20 fistula operations during the last year (*p* = 0.020), and to be affected by low fistula (*p* < 0.001) (Table 2).

These factors resulted as independent variables for the surgeons’ acceptance of fistulotomy also at the multivariate analysis (Table 3).

As declared by the surgeons in this survey, the acceptance to fistulotomy in both low and high anal fistulas was significantly correlated with perspectives of success/failure following the procedure (*p* < 0.001) and the related risk of continence impairment (*p* < 0.001) (Figure 2).

Distributing the respondents on the basis of the continent of origin, the agreement rate to be submitted to the fistulotomy was 69.7% among Middle–South American surgeons, 66.0% for North Americans, 64.4% for Asians, 59.8% for Europeans, 59.1% for Africans, and 45.9% for Oceanian surgeons (*p* = 0.002, Table 4). Comparing the acceptance rate in each continent to that of the rest of the world (Table 4), a significantly higher rate was documented in Middle–South America (69.7% vs. 61.3%, *p* = 0.013), while it was lower among surgeons from Europe (59.8% vs. 64.9%, *p* = 0.041) and Oceania (45.9% vs. 63.5%, *p* = 0.002). Further stratification of the surgeons’ opinion concerning separately low and high fistulas, in each continent and each country are reported in Appendix A.

Moreover, respondents agreeing to be submitted to the fistulotomy showed significantly different pattern of responses to the questionnaire answers compared to those who did not (Figure 3), as follows:−Less frequently would feel the negative impact played by the anal fistula to their quality of life (very much or much in 73.3% vs. 86.7%, *p* < 0.001);−Less frequently would be worried/anxious because of an anal fistula (very much or much in 71.3% vs. 85.7%, *p* < 0.001);−More frequently would consider the perspectives of healing and failure of the anal fistula following fistulotomy as absolutely or enough acceptable (82.3% vs. 53.2%, *p* < 0.001);−Less frequently would be worried about the possibility to be submitted to more than one surgical procedure to treat an anal fistula (very much or much in 63.8% versus 71.2%, *p* < 0.001);−More frequently would consider perspectives about the possible continence impairment as absolutely or enough acceptable (38.4% versus 7.6%, *p* < 0.001).

## 4. Discussion

Fistulotomy is an effective surgical option in anal fistula surgery with high success rates reported in the literature [7,10,11,12,13,14,15,16,17,18,19,20,21,22]. However, when surgeons offer fistulotomy to patients, they could address and discuss several aspects that may influence the treatment strategy. Any surgeon facing to the anal fistula surgery should be conscious and aware of them, and all these aspects (including possibilities of care, risks of recurrence and multiple operations, side effects of surgery, and primarily the risk of continence to feces impairment) should be disclosed.

First, quality of life (QoL) is reduced in anal fistula disease [29,30,31,32]. As reported by respondents, anal fistula could have a very negative impact on QoL in a percentage up to 88.2% of surgeons. Moreover, this effect was greater in case of high fistula or non-acceptance of fistulotomy. In addition, participants were “very much” or “much” worried to be affected by an anal fistula and submitted to more than one surgery (considered as an index of severity and/or unsuccessful treatment for anal fistula disease). In a recent survey analyzing what information the patients with perineal Crohn’s disease would like to know before treatment decisions, the wound and immediate aftercare, the severity of surgical procedure, and its consequences on the anal and perianal area were highlighted as major concerns [33]. Therefore, up to 87.9% of surgeons showed that they would be anxious about fistula disease, in particular if they would affect by a high anal fistula or not accept fistulotomy, probably because they know most of the problems related to this distressing condition and the surgical options adopted in its management [16,34,35].

The knowledge of the healing and failure rates reported in the recent literature on fistulotomy was another nonnegligible topic that could lead respondents to choose it or not. In particular, surgeons were strongly influenced by this aspect in general and in low and high fistulas. Ellis showed that more patients affected by anal fistula would prefer a successful surgical technique (with success rates preferably greater than 30%), even accepting the risk of a degree of anal continence impairment [36]. Indeed, the latter is the other factor that considerably influenced the respondents in this study on the possibility to be submitted to fistulotomy, both in general and in low and high fistulas. Recurrence of fistula disease and impaired anal continence were strictly related to patients’ satisfaction and QoL after fistulotomy [8,36,37,38], even if the consequences of fecal incontinence on patients’ lifestyle emotional stability were devastating [39]. However, QoL significantly improved after an effective fistulotomy, also in patients with a minimal continence deterioration (up to four points in the St. Mark’s continence score) [30].

In the literature, factors associated with postoperative impairment of anal continence were preoperative fecal incontinence, previous anal surgery, gender, and complexity of anal fistula, including a high internal orifice [16,21,22,31]. Interestingly, lower percentages of agreement to underwent fistulotomy were reported and confirmed by multivariate analysis in female surgeons and high fistulas, where the section length of the anal sphincter muscles plays an important role. Although it is widely accepted that the distal third of the anal sphincters could be safely divided, the risk of continence impairment in simple anal fistula remains [11,12,40].

Surgeons’ experience in fistula management was also associated with fistulotomy acceptance. In particular, respondents with less years of experience in anal fistula surgery or more cases treated in the last year would be less willing to undergo fistulotomy. In this case, we could hypothesize that younger surgeons who performed more than 20 surgical procedures for fistula in the last year may prefer other techniques rather than fistulotomy (i.e., the more recently introduced sphincter-sparing procedures), in order to minimize the risk of continence impairment, especially in high fistulas. However, multivariate analysis showed that only surgeons who treated more patients with anal fistula in the last year were less likely to accept fistulotomy.

Taking into consideration all these aspects, surgeons would opt for fistulotomy in 87.5% and 37.8% of patients with low and high fistula, respectively. Moreover, the fistulotomy acceptance rates varied worldwide. Probably, this variability could be the consequence of the different experience, culture, or local guidelines of the surgeons who would agree to be submitted to fistulotomy in case of anal fistula. However, it is possible to speculate that if a surgeon would accept fistulotomy for themself, they would also be inclined to consider and offer this surgical option to the patient. This speculation could be the basis for a healthy doctor–patient relationship based on a greater trust when the surgeon would propose fistulotomy as a treatment for anal fistula.

Each patient with anal fistula disease should be regarded as a unique case and treatment strategy should be individualized considering all the aspects mentioned before. Further studies on patients’ opinions about anal fistula and the possibility to be submitted to fistulotomy may be needed in order to choose the most suitable surgical treatment according to the views of the surgeon and the patient himself.

This study has several limitations. It is widely accepted that online surveys can be subject to bias resulting from the nonrepresentative nature of the web population and self-selection of participants. Generally, the response rate, defined as the proportion of respondents out of the total number who were invited to participate, would provide an accurate idea of the generalizability of the results. However, according to the CHERRIES statement, we decided to use the participation rate as a good tool for the response metrics due to the survey distribution also by social networks [28]. Residents and fellows in colorectal surgery also answered the questionnaire, but they were in a smaller percentage than consultants. Furthermore, respondents showed high experience in fistula management considering their years of practice and the number of cases treated in the last year. Despite the survey being launched around the world, it did not reach an equal geographic distribution, especially in low-income countries. However, anal fistula is a very common disease in Americas and Europe, where colorectal societies and surgeons are particularly sensitive to this disabling condition. Lastly, we are aware that the use of a validated questionnaire could improve understanding and awareness on this topic worldwide.

## 5. Conclusions

Surgeons’ opinions on the adoption of anal fistulotomy are frequently divergent, either in low or high fistulas. However, this study showed that the majority of surgeons recognized the extreme complexity of this approach for both low and high fistulas. Consequences of fistulotomy, including the risk of anal continence impairment, cannot be negligible and, in fact, the majority of surgeons refused to be submitted to fistulotomy as patients. Although this operation remains the most frequently adopted in anal fistula management due to its best healing rate, data of this study highlighted the necessity of an accurate patients’ selection and, if possible, the adoption of alternative strategy (i.e., immediate sphincter repair) to reduce the risk of postoperative impairment of anal continence.

## Figures and Tables

**Figure 1 jcm-12-00825-f001:**
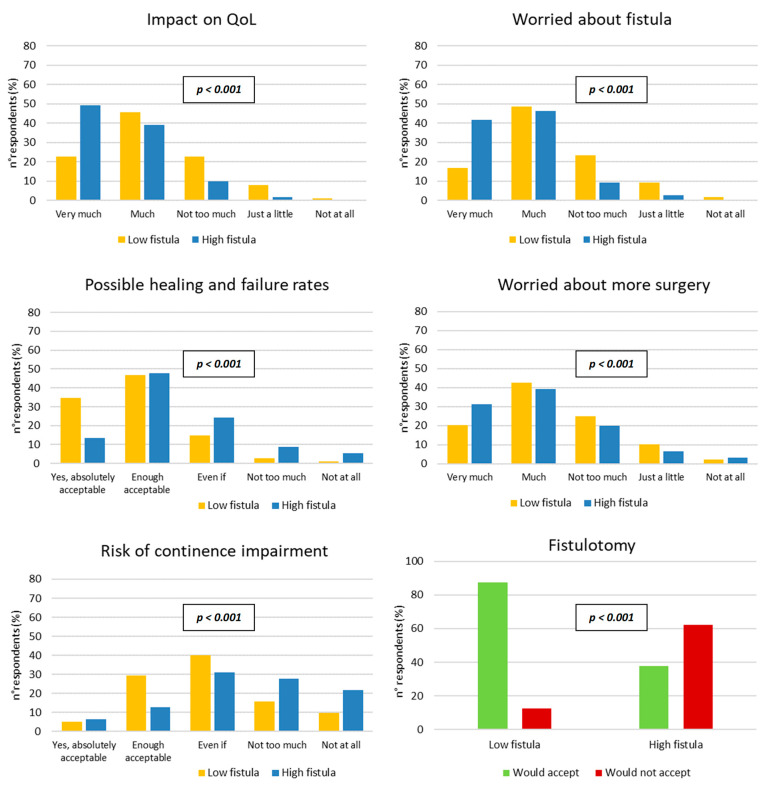
Surgeons’ responses to the questionnaire regarding low and high anal fistulas (chi-squared test). QoL = quality of life; Even if = considerable.

**Figure 2 jcm-12-00825-f002:**
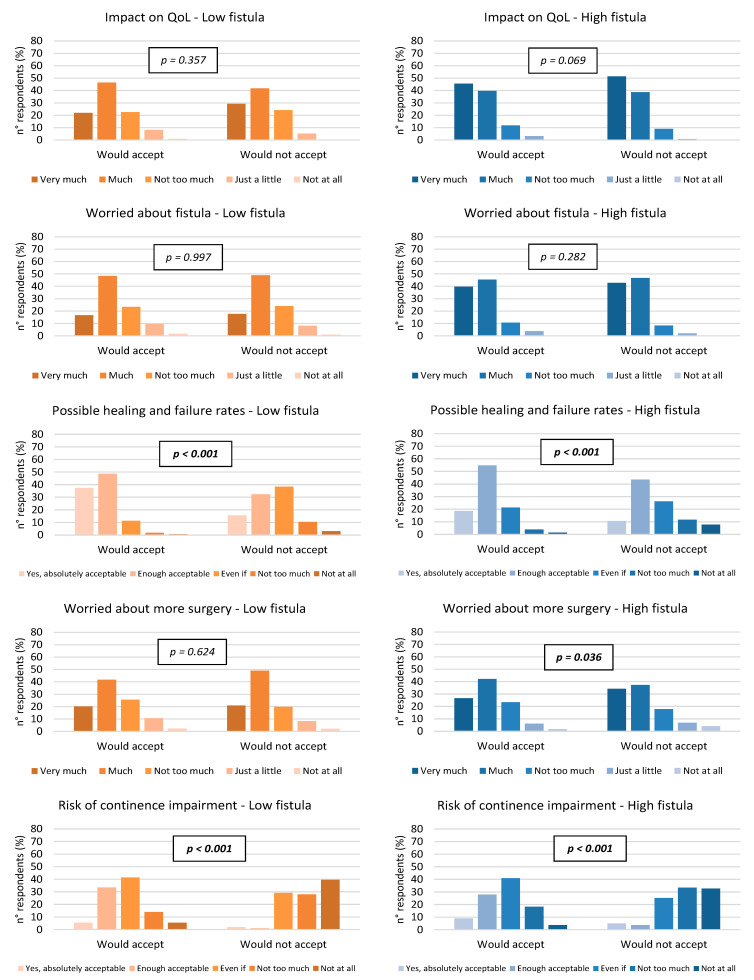
Comparison of respondents’ opinions between who would and would not agree to be submitted to fistulotomy in low and high anal fistula, separately (chi-squared test). QoL = quality of life; Even if = considerable.

**Figure 3 jcm-12-00825-f003:**
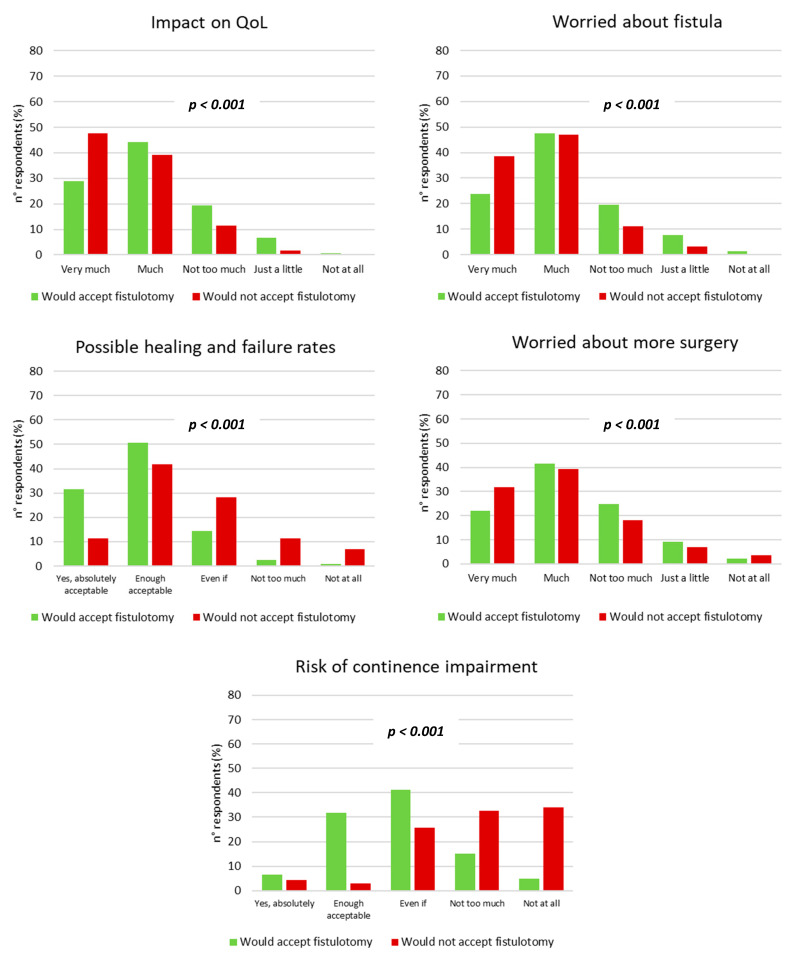
Comparison of respondents’ opinions between who would and would not agree to be submitted to fistulotomy in both low and high anal fistula (chi-squared test). QoL = quality of life; Even if = considerable.

**Table 1 jcm-12-00825-t001:** Baseline characteristics of participating surgeons.

	*n*° Surgeons (%)
Respondents	767
Sex ratio (male:female)	642 (83.7):125 (16.3)
Geographical distribution	
Europe	333 (43.4)
Africa	11 (1.5)
Asia	80 (10.4)
Oceania	37 (4.8)
North America	184 (24.0)
Middle–South America	122 (15.9)
Type of hospital	
Academic	423 (55.1)
Non-academic teaching	209 (27.2)
Non-teaching	135 (17.7)
Training level	
Consultant	683 (89.0)
Resident	38 (5.0)
Fellow	46 (6.0)
Age group (years)	
<30	17 (2.2)
30–39	167 (21.8)
40–49	232 (30.2)
50–59	184 (24.0)
≥60	167 (21.8)
Professional experience in fistula surgery (years)	
0–5	107 (14.0)
6–10	149 (19.4)
11–20	226 (29.5)
>20	285 (37.1)
Professional experience in fistula surgery (*n*° fistula cases treated in the last year)	
None	16 (2.1)
1–10	93 (12.1)
11–20	178 (23.2)
21–30	149 (19.4)
31–40	105 (13.7)
41–50	72 (9.4)
>50	154 (20.1)

**Table 2 jcm-12-00825-t002:** Relationship between respondents’ characteristics and the choice to accept fistulotomy in both low and high fistula (chi-squared test). The *p*-value in bold was statistically significant.

		Would Accept Fistulotomy (%)	*p*-Value
Gender	Male	64.3	**0.003**
	Female	54.4	
Age (years)	<50	61.2	0.195
	>50	64.4	
Type of hospital	Academic	60.8	0.236
	Non-academic teaching	64.8	
	Non-teaching	65.2	
Training level	Consultant	62.3	0.669
	Resident	67.1	
	Fellow	64.1	
Professional experience in fistula surgery (years)	<20	60.8	**0.050**
	>20	65.8	
Professional experience in fistula surgery (*n*° cases in the last year)	<20	66.4	**0.020**
	>20	60.4	
Type of fistula	Low	87.5	**<0.001**
	High	37.8	

**Table 3 jcm-12-00825-t003:** Univariate and multivariate analyses evaluating factors related to the acceptance of fistulotomy in both low and high fistula (binary logistic regression model). The *p*-value in bold was statistically significant.

	Univariate Analysis Relative Risk (95% CI)	*p*-Value	Multivariate Analysis Relative Risk (95% CI)	*p*-Value
Male	1.507 (1.146–1.981)	**0.003**	1.707 (1.220–2.387)	**0.002**
Age <50 years	0.872 (0.708–1.073)	0.195		
Non-teaching hospital	1.198 (0.972–1.476)	0.236		
Resident/fellow	1.148 (0.820–1.607)	0.669		
Professional experience in fistula surgery <20 years	0.806 (0.650–1.001)	**0.050**	0.794 (0.613–1.029)	0.081
Professional experience in fistula surgery <20 cases in the last year	1.293 (1.042–1.605)	**0.020**	1.468 (1.139–1.891)	**0.003**
Low fistula	11.497 (8.874–14.894)	**<0.001**	12.031 (9.245–15.656)	**<0.001**

**Table 4 jcm-12-00825-t004:** Geographical distribution of 767 surgeons who would accept fistulotomy, compared to surgeons who would do that in the rest of the world considering the double response of each single surgeon for low and high fistula together (chi-squared test). The *p*-value in bold was statistically significant.

Geographical Distribution (*n*° Responses)	Would Accept Fistulotomy (%)	Would Accept Fistulotomy in the Rest of the World (%) *	*p*-Value
Europe (666)	398 (59.8)	563 (64.9)	**0.041**
Africa (22)	13 (59.1)	948 (62.7)	0.728
Asia (160)	103 (64.4)	858 (62.4)	0.633
Oceania (74)	34 (45.9)	927 (63.5)	**0.002**
North America (368)	243 (66.0)	718 (61.6)	0.124
Middle–South America (244)	170 (69.7)	791 (61.3)	**0.013**

* In this column, statistical analyses were performed considering data of the overall sample less the answers of respondents from the single continent indicated in the corresponding row.

## Data Availability

The data presented in this study are available on request from the corresponding author.

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
