# Peer review of "Would Surgeons Like to Be Submitted to Anal Fistulotomy? An International Web-Based Survey"

_jcm, 2023, doi:10.3390/jcm12030825_

Round 1
Reviewer 1 Report
This is quite an interesting study in which a large number of colorectal surgeons and trainees were surveyed in regard to a hypothetical scenario in which they were asked to respond as if they were a patient with an anal fistula.
Not surprisingly, the most significant factor influencing the surgeons’ attitudes to fistula management was the level of the fistula, with a high fistula being perceived as more problematic and less likely to be appropriate for treatment with a fistulotomy. In fact it is rather remarkable that nearly 40% of surgeons would accept a fistulotomy for a high fistula.
The authors have defined the survey response rate as the number of surgeons who agreed to participate as a proportion of the number who visited the first survey page, resulting in a response rate of nearly 80%. It would be more appropriate to report the response rate as the proportion of respondents out of the total number who were invited to participate. This is almost certainly a much smaller percentage as is typical in surveys of this type and would provide a more accurate idea of the generalisability of these findings given that there is a strong element of self-selection bias in online surveys, as the authors acknowledge. The final paragraph of the discussion should be modified to reflect the likely true response rate.
The figure legend “even if” as an abbreviation for “Even if high rates of success, the percentages of possible failure are still considerable” should be changed to something which makes sense to the reader – perhaps “considerable”.
The sentence in paragraph 1 of page 12 including “….if a surgeon would accept fistulotomy for himself, he is also inclined…” should be changed to reflect the fact that not all surgeons are male, for example “…..if a surgeon would accept fistulotomy for themself, they would also be inclined…..”
Author Response
Reviewer #1: This is quite an interesting study in which a large number of colorectal surgeons and trainees were surveyed in regard to a hypothetical scenario in which they were asked to respond as if they were a patient with an anal fistula. Not surprisingly, the most significant factor influencing the surgeons’ attitudes to fistula management was the level of the fistula, with a high fistula being perceived as more problematic and less likely to be appropriate for treatment with a fistulotomy. In fact it is rather remarkable that nearly 40% of surgeons would accept a fistulotomy for a high fistula.
Answer: Thanks for your comment, we are glad for that.
The authors have defined the survey response rate as the number of surgeons who agreed to participate as a proportion of the number who visited the first survey page, resulting in a response rate of nearly 80%. It would be more appropriate to report the response rate as the proportion of respondents out of the total number who were invited to participate. This is almost certainly a much smaller percentage as is typical in surveys of this type and would provide a more accurate idea of the generalisability of these findings given that there is a strong element of self-selection bias in online surveys, as the authors acknowledge. The final paragraph of the discussion should be modified to reflect the likely true response rate.
Answer: Thank you for the suggestion. We decided to use the “participation rate” instead of the “response rate” as a tool for the response metrics (as suggested in the CHERRIES statement), because it is impossible to calculate a “real” response rate in an online survey due to its distribution also by social networks (LinkedIn, Twitter, Facebook). Nonetheless, we have modified the Results and Discussion section in order to better clarify this aspect.
The figure legend “even if” as an abbreviation for “Even if high rates of success, the percentages of possible failure are still considerable” should be changed to something which makes sense to the reader – perhaps “considerable”.
Answer: Thanks for your observation. We have changed the legends and figures in order to better explain this aspect.
The sentence in paragraph 1 of page 12 including “….if a surgeon would accept fistulotomy for himself, he is also inclined…” should be changed to reflect the fact that not all surgeons are male, for example “…..if a surgeon would accept fistulotomy for themself, they would also be inclined…..”
Answer: Thank you for the comment. We have changed the text as suggested.
Reviewer 2 Report
This study adresses the topic of patient choice and medical knowledge before fistula surgery. In this inquiry surgeons specialized in ano-rectal surgery were asked their preferences about fistulotomy for anal fistula. This large inquiry included a large number of participants, and results are interesting. It is demonstrated that expertize in fistula surgery influences the "patient's" choice. This reinforces the importance of patient information, and the necessity for alternative surgical options. This topic is pertinent, questions are accurate and methodology original and effective. Redaction is more than perfect.
Author Response
Reviewer #2: This study addresses the topic of patient choice and medical knowledge before fistula surgery. In this inquiry surgeons specialized in ano-rectal surgery were asked their preferences about fistulotomy for anal fistula. This large inquiry included a large number of participants, and results are interesting. It is demonstrated that expertise in fistula surgery influences the "patient's" choice. This reinforces the importance of patient information, and the necessity for alternative surgical options. This topic is pertinent, questions are accurate and methodology original and effective. Redaction is more than perfect.
Answer: Thanks for your comment, we really appreciate it!
Reviewer 3 Report
This study reports the results from an online survey among colorectal surgeons around the world on their opinion and preference about fistulotomy. It is a well written paper with some interesting findings, especially those regarding the surgeon’s age and number of fistula procedures performed over the last year. However, I do have some questions and comments.
The questionnaire used is not validated. To me the most difficult question to interpret is “Would you be worry about the possibility to be submitted to more than one surgery to treat your anal fistula?”. If I prefer a sphincter sparing approach knowing that this likely (or even guarantied) leads to being subjected to more than one operation, is that to worry about the possibility to be submitted to more than one surgery? Or is that not worry about the possibility to be submitted to more than one surgery since I consider that to be a part of a sphincter sparing approach? I believe that this question could be discussed in the discussions section. I also think that using a non-validated questionnaire should be discussed among the limitations of the study.
In my opinion table 4 and table S2 should include both the numbers of surgeons who answered yes and the total number of surgeons who answered the question for each category (e.g. would accept fistulotomy in low fistula in Europe: 278 out of 333) since results from some continents seems underpowered.
Author Response
Reviewer #3: This study reports the results from an online survey among colorectal surgeons around the world on their opinion and preference about fistulotomy. It is a well written paper with some interesting findings, especially those regarding the surgeon’s age and number of fistula procedures performed over the last year.
Answer: Thanks for your comment, we are glad for that.
However, I do have some questions and comments. The questionnaire used is not validated. To me the most difficult question to interpret is “Would you be worry about the possibility to be submitted to more than one surgery to treat your anal fistula?”. If I prefer a sphincter sparing approach knowing that this likely (or even guarantied) leads to being subjected to more than one operation, is that to worried about the possibility to be submitted to more than one surgery? Or is that not worry about the possibility to be submitted to more than one surgery since I consider that to be a part of a sphincter sparing approach? I believe that this question could be discussed in the discussions section. I also think that using a non-validated questionnaire should be discussed among the limitations of the study.
Answer: Thanks for your suggestion. In this case, we considered the possibility to be submitted to more than one surgery as an index of severity and/or unsuccessful treatment for anal fistula disease. Nonetheless, we have added some explanations in the Discussion section, mainly in the limitation of the study.
In my opinion table 4 and table S2 should include both the numbers of surgeons who answered yes and the total number of surgeons who answered the question for each category (e.g. would accept fistulotomy in low fistula in Europe: 278 out of 333) since results from some continents seems underpowered.
Answer: Thanks for your observation, we have modified Tables 4 and S2.